# Antibacterial Properties of Triethoxysilylpropyl Succinic Anhydride Silane (TESPSA) on Titanium Dental Implants

**DOI:** 10.3390/polym12040773

**Published:** 2020-04-01

**Authors:** Judit Buxadera-Palomero, Maria Godoy-Gallardo, Meritxell Molmeneu, Miquel Punset, Francisco Javier Gil

**Affiliations:** 1Biomaterials, Biomechanics and Tissue Engineering Group, Department of Materials Science and Engineering, Technical University of Catalonia (UPC), Av. Eduard Maristany, 10-14, 08019 Barcelona, Spain; judit.buxadera@upc.edu (J.B.-P.); meritxell.molmeneu@upc.edu (M.M.); miquel.punset@upc.edu (M.P.); 2Barcelona Research Centre in Multiscale Science and Engineering, Technical University of Catalonia (UPC), Av. Eduard Maristany, 10-14, 08019 Barcelona, Spain; 3Bioengineering Institute of Technology (BIT), Universitat Internacional de Catalunya (UIC), C. Josep Trueta s/n, 08195 Sant Cugat del Vallès, Spain; mgodoy@uic.es; 4Basic Science Department, Faculty of Medicine and Health Science, Universitat Internacional de Catalunya (UIC), C. Josep Trueta s/n, 08195 Sant Cugat del Vallès, Spain; 5UPC Innovation and Technology Center (CIT-UPC), Technical University of Catalonia (UPC), C. Jordi Girona 3-1, 08034 Barcelona, Spain; 6School of Dentistry, Faculty of Medicine and Health Science, Universitat Internacional de Catalunya (UIC), C. Josep Trueta s/n, 08195 Sant Cugat del Vallès, Spain

**Keywords:** titanium, triethoxysilylpropyl succinic anhydride (TESPSA), bacterial adhesion, antibacterial, biofilm, silane

## Abstract

Infections related to dental implants are a common complication that can ultimately lead to implant failure, and thereby carries significant health and economic costs. In order to ward off these infections, this paper explores the immobilization of triethoxysilylpropyl succinic anhydride (TESPSA, TSP) silane onto dental implants, and the interaction of two distinct monospecies biofilms and an oral plaque with the coated titanium samples. To this end, titanium disks from prior machining were first activated by a NaOH treatment and further functionalized with TESPSA silane. A porous sodium titanate surface was observed by scanning electron microscopy and X-ray photoelectron spectroscopy analyses confirmed the presence of TESPSA on the titanium samples (8.4% for Ti–N-TSP). Furthermore, a lactate dehydrogenase assay concluded that TESPSA did not have a negative effect on the viability of human fibroblasts. Importantly, the in vitro effect of modified surfaces against *Streptococcus sanguinis, Lactobacillus salivarius* and oral plaque were studied using a viable bacterial adhesion assay. A significant reduction was achieved in all cases but, as expected, with different effectiveness against simple mono-species biofilm (ratio dead/live of 0.4) and complete oral biofilm (ratio dead/live of 0.6). Nevertheless, this approach holds a great potential to provide dental implants with antimicrobial properties.

## 1. Introduction

The oral cavity is the main gateway to the human body and hosts the second largest microbiome after the gut, showing great diversity with more than 770 microbial species described [1,2]. Coevolution and mutual adaptation have led to a finely-tuned ecosystem, which not only inhibits colonization by pathogenic microorganisms, but also aids the maturation of the host’s immune system. However, upon disruption of the balance between host and the oral microbiota, pathogenic bacteria can cause periodontitis and tooth decay (caries) – the two main dental oral diseases that may lead to teeth loss [3]. 

Among the wide range of available biomaterials, titanium and its alloys are the most widespread material in dentistry for the restoration of dental function. Its excellent biocompatibility, high corrosion resistance, and low density, renders these metals close to the mechanical properties of bones [4]. However, despite its high success rate, lack of osseointegration after early healing, or bacterial infection of the peri-implant tissue, can lead to device failure [5,6,7]. For example, pathogenic biofilm formation on the surface of dental implants may result in peri-implantitis, a site specific and highly destructive inflammatory process in the peri-implant mucosa. This affects both the soft and hard tissue surrounding the dental implant [8], weakening the connection between the bone and the implant and ultimately leading to implant loss [9,10,11].

Consequently, there is a strong need to develop new strategies to combat biofilm-related implant infections, e.g., by functionalizing the surface of dental implants with materials that are unfavourable for biofilm attachment [12,13], and thereby improving the long-term success rate.

During the last decades, systemic antibiotic prophylaxis was the treatment of choice for patients after medical dental implantation to prevent postsurgical infections [14,15]. However, routine antibiotic treatments are usually unable to prevent implant-associated infections where bacteria reached a state of maturity as biofilm [16,17]. Biofilms are hardly accessible for drugs and the host defense system due to its secreted protective layer consisting of extracellular DNA, proteins, and polysaccharides. In addition, the steadily increasing use of antibiotics in recent decades have led to the emergence of antibiotic resistant bacteria (ARB) [18,19]. Importantly, according to a review commissioned by the United Kingdom government, antimicrobial resistance (AMR) will lead to the death of 10 million people a year by 2050 [20]. In consequence, there is an inevitable need to develop new antibacterial strategies to minimize the inappropriate use of antibiotics.

Over the last years, various therapeutic strategies have been proposed to combat bacterial infections related to dental implants [12,21,22], such as the covalent immobilization of antibacterial molecules onto the surface to obtain a stable bioactive coating [23,24,25]. For this purpose, self-assembled monolayers (SAMs) of organic molecules may serve as an anchor for subsequent attachment of biomolecules [26,27,28,29]. In the case of silanization, silane can form a covalent bond with pre-activated titanium, followed by the covalent immobilization of biomolecules that are able to trigger various cell responses. This effect can be further improved by using silanes that harbour biological activities themselves. For instance, the silane triethoxysilylpropyl succinic anhydride (TESPSA) has been shown to present both osteoinductive and antibacterial activity when immobilized on titanium surfaces [30,31]. Importantly, the biological activity of the silane and its straightforward chemistry allows for short and simple protocols, rendering this type of functionalization especially feasible for applications in the dental industry. 

Here, we report the successful immobilization of TESPSA silane onto a commercially available dental implant. Specifically, we: (i) studied the possibility of applying this treatment to a dental implant, (ii) characterized the treated surface to confirm successful incorporation, (iii) verified the absence of cytotoxicity towards human fibroblasts, and (iv) studied the antibacterial activity in both single and multiple-species models of oral biofilm.

## 2. Materials and Methods 

### 2.1. Materials and Reagents

Sodium hydroxide (NaOH), triethoxysilylpropyl succinic anhydride silane (TESPSA), toluene, N,N-diisopropylethylamine (DIEA), isopropanol, ethanol, acetone and paraformaldehyde were purchased from Sigma-Aldrich (St. Louis, MO, USA). Dulbecco’s modified Eagle’s medium (DMEM), phosphate buffered saline (PBS), fetal bovine serum (FBS), trypsin, L-glutamine, penicillin/streptomycin, palloidin-rhodamin, 4′,6-diamidino-2-phenylindole (DAPI) and LIVE/DEAD BacLight bacterial viability kit were purchased from Invitrogen (Carlsbad, CA, USA).

Cytotoxicity Detection Kit LDH was purchased from Roche Applied Science (Penzberg, Germany). Mammalian extraction reagent (M-PER^®^) was purchased from Pierce (Waltham, MA, USA). *Streptococcus sanguinis* was purchased from Colección Española de Cultivos Tipo (CECT, CECT 480, Valencia, Spain). *Lactobacillus salivarius* was purchased from the University of Göteborg Culture Collection (CCUG, CCUG 17826, Göteborg, Sweden). Todd-Hewitt (TH) broth and Man-Rogosa-Sharpe (MRS) broth were purchased from Scharlau (Scharlab SL, Sentmenat, Spain). Brain Heart Infusion Broth (BHI) was purchased from Difco (Detroit, MI, USA). The solutions were made with ultrapure water (Millipore Milli-Q, Merck Corporation, Kenilworth, NJ, USA). 

### 2.2. Sample Preparation

Titanium rods prior machining, as used in the production of dental implants, were cut into disks of 5 mm diameter and 3 mm thickness. The metallic surfaces were immersed in a NaOH solution (5 M, 500 µL) at 60 °C for 24 h and washed in ultrapure water (30 min twice), methanol, acetone and dried with hot air. Next, the treated samples were immersed in a solution of silane triethoxysilylpropyl succinic anhydride (TESPSA, 0.5% v/v in anhydrous toluene) for 1 h at 70 °C in absence of oxygen. A basic media was achieved by adding a dissolution of N,N-diisopropylethylamine (DIEA 3 % (v/v)). After washing 10 min by sonication in toluene, the titanium samples were washed with isopropanol, ethanol, distilled water, acetone and dried with nitrogen (see a summary of the strategy in Figure 1). Prior the assays, all materials were treated with gamma radiation to sterilize them [11,12,13].

### 2.3. Physico-Chemical Characterzization

Topography and surface microstructure of the implants were analyzed by scanning electron microscopy (SEM, Zeiss Neon40 FE-SEM, Carl Zeiss NTS GmbH, Oberkochen, Germany). 

Roughness was studied with a white-light optical profilometer (Wyko NT9300 Optical Profiler, Veeco Instruments, Plainview, NY, USA), in the vertical scanning interferometry mode with a 5× objective, resulting and a scanning area of 736 µm × 480 µm. Data was analyzed with the Veeco Vision 4.10 software, to obtain the arithmetic average height (R_a_).

Measurements of the contact angle in the static mode were used to calculate the wettability and surface energy of the surface, in the sessile drop method (Contact Angle System OCA15 plus; Dataphysics, Filderstadt, Germany). The tests were done with ultrapure distilled water and diiodomethane. The surface energy was calculated according to the Owens, Wendt, Rabel and Kaelble (OWRK) equation [32]:(1)γL(1+cosθ)=2((γLdγSd)12+(γLpγSp)12),

In the OWRK equation, γ_L_ is the solid surface energy, and γ^d^_L_ and γ^p^_L_ are respectively the dispersive and the polar component of the liquid surface tension. Θ is the measured contact angle (CA) and 3 µL drop at a dosing rate of 1 µL min^−1^ was used. The tests were done at 25 °C, and the results were processed with SCA 20 software (Dataphysics). Four measurements were performed in four different samples.

X-ray photoelectron spectroscopy (XPS) was used to study the surface chemical composition, at 150 W and a pressure below 7.5 × 10^−9^ mbar with a Phoibos 150 MCD-9 detector (D8 advance, SPECS Surface Nano Analysis GmbH, Berlin, Germany). A pass energy of 25 eV and 0.1 eV steps were used to record high resolution spectra was recorded. Three samples for each condition were analyzed by referring the binding energies to the C1s signal. 

Stability of the coating was assessed by immersing the samples in phosphate buffered saline (PBS) for 1 and 4 days at 37 °C. After each time point, samples were rinsed with ultrapure distilled water, dried with nitrogen and the chemical composition was assessed by XPS.

### 2.4. In Vitro Cell Assays

Human foreskin fibroblasts (HFFs) between passage three and six were cultured in supplemented DMEM (supplemented with % FBS, 1% (w/v) L-glutamine and 1% penicillin/streptomycin). Cultures were maintained in a humidified incubator at 37 °C and 5% CO_2_, and the medium was renewed every 2 days. Detachment of HFFs cells was performed by incubation in trypsin during 5 min, centrifugation at 300 g for 5 min and resuspension in fresh supplemented medium.

#### 2.4.1. Cell Cytotoxicity Assay

Indirect cytotoxicity was performed according to ISO 10993-5, by incubating the samples at 37 ± 2 °C for 72 h and diluting the extracts at five different concentrations: 100%, 50%, 10%, 1% and 0.1%. HFFs were seeded at a density of 5 × 10^3^ cells per well in a 96-well plate and incubated with complete medium for 24 h. After incubation time, the culture media was replaced with the extracts of the samples for further 24 h. Cells were then lysed with 200 µL of M-PER^®^ per well. Cell viability was quantified by the Cytotoxicity Detection Kit LDH, which measures the released lactate dehydrogenase (LDH). The measurements were performed by the absorbance measurement at 490 nm with a spectrophotometer ELx800 Universal Microplate Reader (Bio-Tek Instruments, Inc. Winoosk, VT, USA). Medium without cells in the well was the negative control and lysed cells cultured on TCPS were the positive control. For every measurement, the cytotoxicity was calculated as: % Cytotoxicity = (experimental result − negative control)/ (positive control-negative control). Triplicates of each condition were evaluated.

#### 2.4.2. Cell Adhesion Assay

Cell adhesion was studied by fluorescence cell imaging. 2 × 10^4^ HFFs cells were seeded onto titanium samples and were incubated for 24 and 48 h at 37 °C and 5% CO_2_. The cells were then rinsed 3× with PBS (200 µL) and fixed with a solution of paraformaldehyde (4% in PBS) followed by 3× washes in PBS. For staining, cells were permeabilized in a solution of Triton X-100 (0.05% in PBS) for 20 min. After 3× washing in PBS, the actin filaments were stained by incubating the samples in a solution of palloidin-rhodamin (0.1 µg mL^−1^) in PBS for 1 h. Samples were then washed 3× with PBS and immersed in a solution of 4′,6-diamidino-2-phenylindole (DAPI, 20 µg mL^−1^) for 5 min in order to nuclei observation. Disks were washed 3× in PBS and images were taken with a confocal laser scanning microscope (Leica TCS-SP5 Microscope, Leica Microsystems, Wetzlar, Germany) with the excitation/emission wavelengths of 405/440–480 nm and 543/55–595 nm. Five images of each sample were taken.

### 2.5. Bacterial Strains and Culture Conditions

Bacterial assays were performed with *Streptococcus sanguinis* (*S. sanguinis*) and *Lactobacillus salivarius* (*L. salivarius*). Besides, dental plaque collected from one volunteer was also studied. *S. sanguinis* was grown and maintained in Todd-Hewitt broth (TH), *L. salivarius* in Man-Rogosa-Sharpe broth (MRS) and oral plaque in Brain Heart Infusion Broth (BHI) (Difco). Cultures were incubated overnight at 37 °C before each assay in the appropriate culture medium. After this time, the suspension optical density of the single strain culture was adjusted to 0.2 ± 0.01 at 600 nm, giving approximately 1 × 10^8^ colony forming units (CFU) mL^−1^. In the case of oral plaque, the optical density was adjusted to 0.1 at 405 nm and then diluted 1:10. All assays were performed in static conditions, and three replicates for each working condition were used.

#### 2.5.1. Bacterial Adhesion onto Functionalized Samples

Samples in a 24-well plate were incubated with 1 mL of bacterial suspension during 2 h at 37 °C. After this time, the non-adhered bacteria were removed by washing twice with PBS. In order to harvest the adhered bacteria, each sample was vortexed in 1 mL of PBS for 5 min. The collected bacteria were then seeded on agar plates by serial dilutions and incubated at 37 °C during 24 h, in order to quantify the resulting colonies. 

#### 2.5.2. Viability of Bacteria onto Modified Samples

In order to observe the bacteria viability on the samples, a LIVE/DEAD BacLight bacterial viability kit was used. This assay is based on the use of two DNA staining agents. Syto 9, which shows green fluorescence, is able to stain all the cells, and propidium iodide, which reveals red fluorescence, is only able to penetrate damaged membranes and shows red fluorescence. Samples were immersed in 1 mL of the bacterial suspension (1 × 10^8^ cells mL^−1^) and incubated anaerobically at 37 °C for different time points (1, 2 and 4 weeks). At each time, samples were rinsed 2× with PBS. The staining was done by immersing the samples in 50 µL of the two dyes in NaCl (1.5 µL of SYTO^®^ and 1.5 µL of propidium iodide in 0.85% NaCl solution) at room temperature and protected from light for 15 min. The attached bacteria were observed by confocal laser scanning microscopy (CLSM), with the software EZ-C1 v3.40 build 691(3.9, Nikon, Tokyo, Japan), with a 20× objective. Five images of five random positions of the surfaces were acquired, and a stack of 40 slices (each 1 µm thick each) was scanned. The quantification of the images was performed with Imaris software (Belfast, United Kingdom), by calculating the volume ratio of red fluorescence (cells with damaged membrane) vs. green and red fluorescence (total cells). Triplicates of each condition were analyzed:(2)$$\mathrm{volume}\;\mathrm{ratio}\;\mathrm{of}\;\mathrm{dead}\;\mathrm{cells}=\frac{\mathrm{volume}\;\mathrm{of}\;\mathrm{red}\;\mathrm{bacterial}}{\mathrm{volume}\;\mathrm{of}\;\mathrm{red}\;\mathrm{bacteria}+\mathrm{volume}\;\mathrm{of}\;\mathrm{green}\;\mathrm{bacteria}}\;,$$

### 2.6. Statistical Analysis

Data was analyzed by a U Mann-Whitney test using the IBM SPSS Statistics 20 software (20.0, IBM, Armonk, NY, USA). *p* < 0.05 was the used significance level. 

## 3. Results

### 3.1. Physico-Chemical Characterization

Non-activated titanium disks (Ti) showed the typical grit-blasting topography (Figure 2a(i)), while the samples treated with sodium hydroxide (Ti–N) exhibited a sodium titanate microporous-structure with a representative pore diameter around 1 µm (Figure 2a(ii)) [18,19,20]. Moreover, silanization with TESPSA did not reveal any effect onto the modified titanium morphology (data not shown). The values of Ra parameter are displayed in Figure 2b. Ti–N samples presented a statistically significant increase in roughness compared to Ti control. The TESPSA coating did not modify the surface roughness compared to Ti–N.

Ti–N–TSP samples presented a higher wettability compared to the Ti control surfaces (Table 1), as shown by the lower water contact angle values (CA), as well as the higher polar component (POL) of the surface free energy (SFE). The dispersive component (DISP) of the SFE revealed a lower contribution for both Ti and Ti–N–TSP conditions. 

The atomic composition of Ti and Ti–N–TSP was evaluated by means of X-ray photoelectron microscopy (XPS) (Figure 3a). The covalent attachment of TESPSA onto titanium disks slightly increase in the percentage of carbon (58.5%) but also in a significant increase of silicon presence (8.4%). The increase of the silicon presence on the surface indicates the success of the silanization process (1.7% for Ti and 8.4% for Ti–N–TSP) which is also confirmed with the Si/Ti ratio (0.25 for Ti and 9.3 for Ti–N–TSP). Control titanium presented four C 1s peaks (284.8, 286.3, 288.2 and 289.0) which can be attributed to hydrocarbon contamination (C–H, C–O and C=O bonds) (Figure 3b(i),c). After TESPSA immobilization, an increase in the silicon signal (Si–O) (Figure 3b(ii),c) and in the C 1s peak were measured. These results can be explained by the TESPSA presence.

The stability of the TESPSA coating was assessed by XPS after immersion in PBS for 1 and 4 days. The Table 2 did no shows statistically differences in the atomic percentages of the three study conditions. However, the slight increase in O and Ti could be related to an oxidation increase (TiO_2_).

### 3.2. Cell Cytotoxicity Assay

Once the covalent attachment of TESPSA onto the titanium surface was established, the viability of the human foreskin fibroblasts (HFFs) cells exposed to the treated samples was studied. The percentage of cell viability after 24 h of incubation in different dilutions is shown in Figure 4. HFFs indirect cytotoxicity assay did not reveal toxicity for any of the studied conditions, since all the cells showed less than 20% reduction in viable cells for all evaluated extracts.

We assessed the cell adhesion of HFFs onto Ti control and Ti–N–TSP by confocal laser scanning microsopy (CLSM) (Figure 4b). As expected, cell adhesion was observed in both non- and treated surfaces but a higher amount of cells after 48 h could be observed onto TESPSA-modified sample. 

### 3.3. Bacterial Adhesion onto Functionalized Samples

TESPSA coated samples displayed a reduction in bacterial adhesion (Table 3) for *S. sanguinis*, *L. salivarius* and oral plaque after 2 h of incubation. The highest reduction was observed for *S. sanguinis* which confirms the different effectiveness of antibacterial coatings among distinct bacterial strains. Moreover, differences were observed between single and multiple-species which corroborates the importance of using a complete oral biofilm model for testing antibacterial properties of antimicrobial coatings.

### 3.4. Viability of Bacteria onto Modified Samples

Analysis of bacteria viability on the samples after 1, 2 and 4 weeks (Table 4) showed an important decrease on the bacteria number on Ti–N–TSP samples compared to Ti control, in single strain assay with *S. sanguinis* and *L. salivarius* and oral plaque. An important bactericidal effect was observed when the titanium was treated with TESPSA (Ti–N–TSP). It should be noted that the single-bacteria biofilm (*S. sanguinis* and *L. salivarius*) revealed a lower viability compared to oral plaque.

In order to confirm the bactericide effect of the Ti–N–TSP samples, an analysis of the ratio of dead/life bacteria was calculated (Table 5). The results revealed that modified sample exhibited a higher ratio than titanium surfaces through all studied time points. Moreover, while the different biofilms showed no differences for the first two weeks of incubation, after four weeks, a higher amount of dead bacteria were observed.

Figure 5 displays viable cells (green dots) and dead cells (red dots) obtained by confocal laser scanning microscope (CLSM). TESPSA clearly reduced the amount of viable bacterial cells while the effectiveness of the treated surface varies depending on the mono-species or oral plaque.

## 4. Discussion

This work demonstrates the efficiency of TESPSA to decrease bacterial adhesion onto titanium dental implants. Importantly, titanium-based implants have been used for long time and are considered a safe dental therapy to replace missing teeth. Despite remarkable progress in biomaterials for dental devices [33], the risk of infection and early implant failure is still a challenge [34]. Thus, the strategy outlined in this study shows a great potential for clinic applications.

As discussed above, peri-implantitis is a progressive and irreversible multi-bacterial infection that may lead to rapid bone loss around the dental implant [35,36]. Therefore, the inhibition of microbial adhesion and thereby biofilm formation represents a potential strategy to reduce or suppress implant failure. Consequently, a surface with antibacterial properties but good cell adhesion could be an attractive way to elude bacteria adherence and improve osseointegration. Organosilanes containing alkyl groups are exceptionally useful to modify the properties and chemical functions on hydroxyl-terminated surfaces by immobilization of biomolecules [37]. Previous studies explored the use of distinct silanes in order to covalently attach antibacterial peptides onto metallic surfaces [38,39,40]. They could verify the antibacterial properties of their materials but which fully depended on the immobilized peptides. However, Godoy-Gallardo et al. [30,41] demonstrated that TESPSA silane is able to provide antibacterial behavior to devices without the presence of any additional biomolecule. In this project, we tested the theory that TESPSA silane can impart antibacterial properties to dental implants upon covalent attachment to the surface. Please see Figure 1 for a schematic of the applied strategy.

In order to provide hydroxyl groups onto dental implants and to ensure optimal silanization, the TESPSA immobilization protocol started with an activation of the samples by alkaline etching [42]. Scanning electron microscopy (SEM) displayed a stable amorphous titanate layer with a nanoporous surface morphology (Ti–N) originating from the activation process, as reported previously [43,44,45] (Figure 2a(ii)). However, the TESPSA immobilization onto the activated surface (Ti–N–TSP) did not reveal any additional differences in the SEM analysis (image not shown). Notably, surface characteristics such as roughness, hydrophilicity and chemical composition affect the adhesion of the cells and bacteria. In particular, the initial attachment of bacteria on rough surfaces are assisted by surface irregularities (i.e., pits and grooves) that may protect the adhered microbial from the environment and allows the bacteria to attach to more areas of the substratum [46]. Consistent with the morphology results, an increase in the surface roughness was observed for NaOH-etched samples (Ra for Ti 0.4 µm; Ra for Ti–N 2.2 µm; Ra for Ti–N–TSP 2.5 µm; Figure 2b). In this work, the values of roughness are higher than 0.2 µm [47,48] which means a high accumulation of bacteria on the surface.

As previously commented, cell and bacterial adhesion is also related with the wettability and surface free energy (SFE) due to surface exposed hydrophobic or hydrophilic chemical groups [49,50,51]. As shown in Table 2, contact angle values of Ti–N–TSP decreased compared to Ti. Despite the hydrophobic character of TESPSA, this effect is attributed to the presence of residual hydroxyl groups on the surface after the NaOH treatment. Consistent with this result is the increase of the polar component of the SFE. Importantly, it has to be mentioned that materials with SFE values above 30 mJ m^−2^ promote optimal cell adhesion values [52]. In this respect, the determined SFE value for the Ti–N–TSP sample is 78 mJ m^−2^, therefore we expect that the wettability of the TESPSA-coated sample will not compromise cell adhesion.

The success of TESPSA immobilization was further quantified by means of XPS analysis. Notably, TESPSA silanization was followed by a drastic increase in the high resolution signal of Si which is linked to the presence of silicon, a key characteristic of silanes (Figure 3a). Moreover, O 1s XPS revealed a peak at a binding energy of 532.5 eV which originates from the Si–O bond [53] (Figure 3b(ii),c). Additionally, we tested TESPSA stability by PBS immersion for 1 and 4 days (Table 2). XPS revealed no significant differences in the high-resolution signal of C 1s, O 1s and Si 2p before and after PBS immersion times. This confirmed TESPSA stability and that no degradation from the titanium surface is happening. 

Upon successfully establishing the functionalization protocol, we tested the effect of TESPSA-coated implants on HFFs cells by an indirect cell viability assay. As we expected, TESPSA did not showed toxic effects against HFFs [30,40,41]. Our results revealed a maximum reduction of 15% in cell viability when all or half-diluted medium extract was used. This is well below the threshold defined by the International Organization for Standardization [54] for the in vitro cytotoxicity of medical devices. In a next step, to test for cell adhesion, we allowed HFF cells to attach to the titanium surface for 24 and 48 h, and subsequently assessed their adherence by confocal laser scanning microscopy (CLSM) (Figure 4b). For imaging, nuclei were labeled with DAPI (blue), and actin filaments with phalloidin-rhodamin (red). As expected, cells showed spread morphology with well-defined cytoskeleton in all conditions. Based on these encouraging results, we focused on the antibacterial properties of the TESPSA-coated disks in the next step.

The oral biofilm consists of a complex bacterial community initiated by free-floating cells adhering to the implant surface. Early colonizers such as *S. sanguinis* are important in that process, due to their role in the guidance of later colonizers [55,56]. Similarly, species like *L. salivarius* are important because their by-products are crucial for biofilm formation and maintenance, but also due to their interplay with other species [57]. 

It is well described in the literature, that the efficacy of specific treatments depends on the mono-species biofilms tested. Importantly, oral biofilms are built up by more than 700 different bacterial species, and testing regimes have to resemble them as closely as possible to reflect the in vivo situation of dental implants. Therefore, in this study we have tested our functionalized titanium samples not only with two distinct mono-species biofilms but also with a complete oral biofilm. TESPSA-coated surfaces revealed a high potential in reducing bacterial adhesion (Table 3), which also confirmed successful coating. The observed differences between mono-species and oral plaque highlight the importance of using more than one biofilm model, and ideally including also multi-species one. 

Finally, to verify that the treated-samples retain their antibacterial properties also in the long-term, the experiments were prolonged for up to 4 weeks (Table 4 and Table 5 and Figure 5). The results showed that even after 4 weeks of incubation, TESPSA-coated disks showed a significant reduction in bacterial viability. However, differences among all three tested biofilms were observed, intriguingly exhibiting even a higher antibacterial effect against the tested oral biofilm. Importantly, this trend is in line with our bacterial adhesion experiments.

Taking together, this study demonstrated that TESPSA holds a great potential for grafting antibacterial properties onto the surface of dental implants. Moreover, our results are of great interest as they highlight the importance of using multi-species models for testing in vitro antibacterial behaviour. Finally, with this study we laid the foundation for further research to explore the use of additional biomolecules grafted onto TESPSA-modified surfaces to promote supplementary cell responses that can further improve osseointegration.

## 5. Conclusions

In this study, we present the characterization of a new silanization treatment of titanium-based dental implants. We anchored TESPSA onto the implants surface to study its bactericidal behavior and to test its biocompatibility with human fibroblasts. The modified surfaces changed the topography of the titanium to a microporous layer of sodium titanate (Ra of 0.4 µm for Ti and 2.5 µm for Ti–N–TSP). This causes a significant decrease in contact angle (from 67.9 to 38.7°) while increasing the surface free energy (45.5 to 78.1 mJ m^−2^). We could show a significant inhibition of bacterial adhesion and biofilm formation in both bacterial single-species (*S. sanguinis* and *L. salivarius*) biofilm models; e.g., we identified dead/live ratios of 0.39 and 0.47 after 4 weeks of incubation, respectively. Importantly, at the same timepoint an even stronger bactericidal effect could be shown in the in vitro oral plaque model (dead/live ratio of 0.58). This is of special interest as the multi-species biofilm model is much more similar to the microbiology involved in peri-implantitis. Ultimately, our results highlight the great potential of TESPSA-coating for a variety of dental applications.

## Figures and Tables

**Figure 1 polymers-12-00773-f001:**
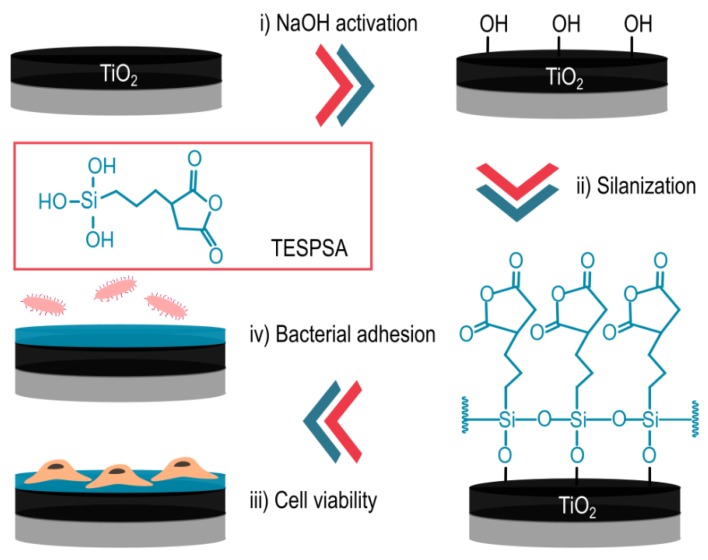
Functionalization strategy of triethoxysilylpropyl succinic anhydride (TESPSA). (**i**) Titanium surface (TiO_2_) is activated by NaOH treatment. (**ii**) Next, the activated surface is coated with TESPSA molecules by silanization process. (**iii**) Treated surfaces did not reveal toxic effects onto fibroblast cells. (**iv**) Following cell viability assay, TESPSA surfaces are able to reduce bacterial adhesion.

**Figure 2 polymers-12-00773-f002:**
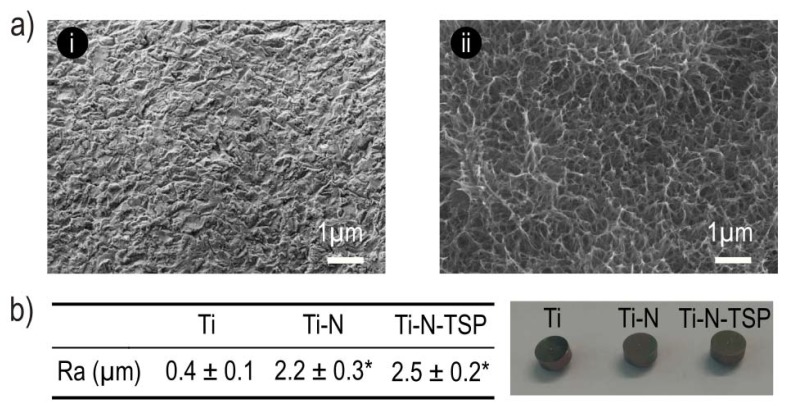
(**a**) Representative scanning electron microscopy (SEM) images of titanium dental implants (i) non-activated titanium dental implant (Ti) and (ii) titanium dental implant thermochemical treated with NaOH 5M (24 h treatment, 60 °C) (Ti–N). (**b**) Ra values for each treatment and samples images. * indicates statistically significant differences compared to Ti (*p* < 0.05).

**Figure 3 polymers-12-00773-f003:**
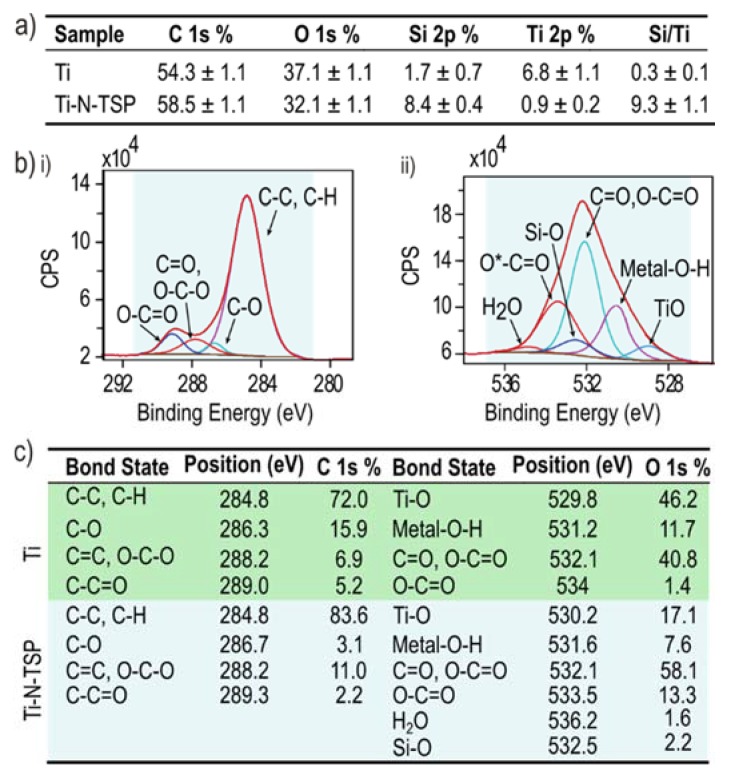
(**a**) Surface’s chemical composition (atomic percentage, %) and Si/Ti relative atomic ratio (mean ± standard deviation (SD)). (**b**) Deconvolution of high resolution spectra of (i) carbon and (ii) oxygen. (**c**) Peak position (eV) and relative intensities (percentages, %) of O 1s and C 1s signals for Ti and Ti–N–TSP samples (mean ± standard deviation (SD)).

**Figure 4 polymers-12-00773-f004:**
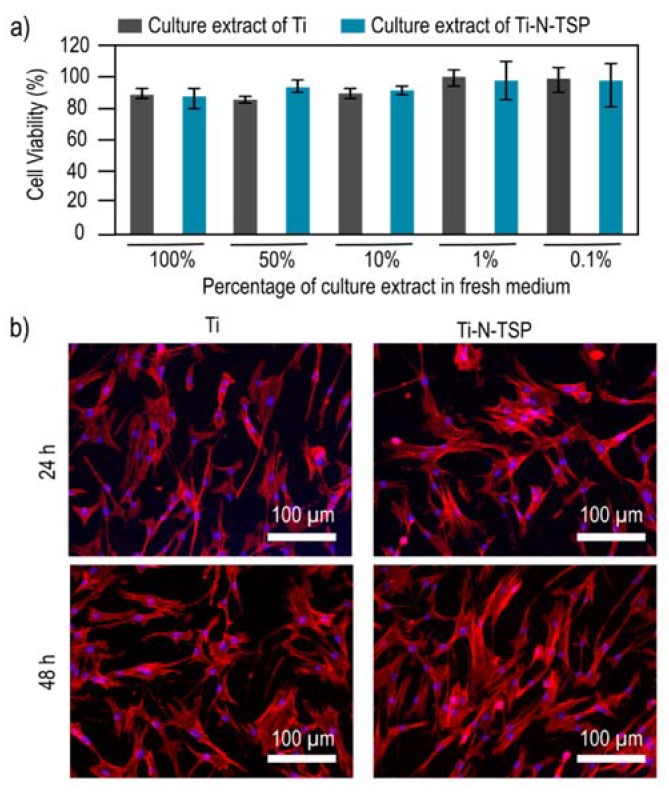
(**a**) Cell viability of Human Foreskin Fibroblasts (HFFs) cells onto titanium surfaces after 1 day of incubation at different culture extract concentrations. (**b**) Confocal laser scanning microscopy (CLSM) images of HFFs cells adhesion at 24 and 48 h. Phalloidin−rhodamine (red signal) was used to stain the actin filaments of the cells. The blue fluorescence signal arises from 4′,6-diamidino-2-phenylindole (DAPI), nucleus staining.

**Figure 5 polymers-12-00773-f005:**
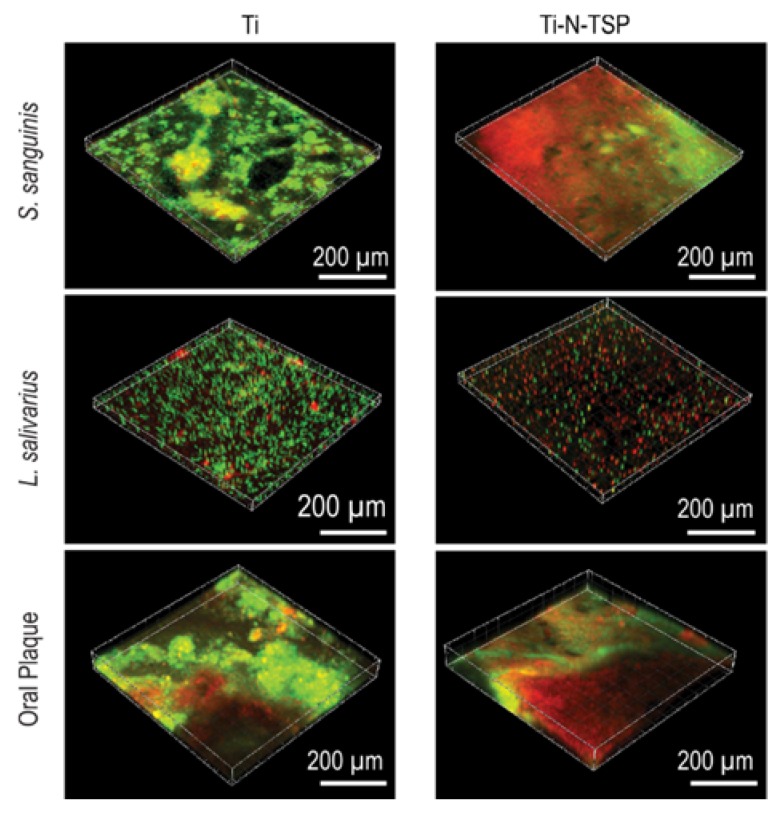
Live (green)/dead (red) staining of *Streptococcus sanguinis*, *Lactobacillus salivarius* and complete oral plaque incubated at 37 °C after 4 weeks of incubation onto titanium (Ti) and TESPSA modified (Ti–N–TSP) samples.

**Table 1 polymers-12-00773-t001:** Water contact angle (CA), surface free energy (SFE) and its dispersive (DISP) and polar (POL) components (mean ± standard deviation (SD)). * indicates statistically significant differences compared to Ti (*p* < 0.05).

Sample	CA (°)	SFE (mJ m^−2^)	DISP (mJ m^−2^)	POL (mJ m^−2^)
Ti	67.9 ± 3.7	45.5 ± 1.9	40.4 ± 3.2	8.7 ± 1.5
Ti–N-TSP	38.7 ± 5.1*	78.1 ± 4.1*	48.6 ± 1.7*	22.9 ± 2.1*

**Table 2 polymers-12-00773-t002:** Chemical composition (atomic percentage (%), mean ± standard deviation (SD)).

Sample	C 1s	O 1s	Si 2p	Ti 2p
Ti–N–TSP	69.1 ± 2.1	25.2 ± 1.5	4.9 ±1.0	0.8 ± 0.1
Immersion 1 day	66.5 ± 3.1	26.9 ± 2.3	5.7 ± 0.9	0.9 ± 0.2
Immersion 4 days	64.7 ± 2.3	27.2 ± 2.5	6.9 ± 1.2	1.2 ± 0.4

**Table 3 polymers-12-00773-t003:** Value of the colony-forming unit (CFU) cm^−2^ of *S. sanguinis*, *L. salivarius* and oral plaque after 2 h of incubation on titanium (Ti) and TESPSA-modified (Ti–N–TSP) samples. Asterisk means statistical significant differences (*p* < 0.05). (mean ± standard deviation (SD)).

Sample	*S. Sanguinis*	*L. Salivarius*	Oral Plaque
Ti	150 ± 21^*^	1198 ± 120^*^	1850 ± 78^*^
Ti–N–TSP	12 ± 7	450 ± 33	591 ± 22

**Table 4 polymers-12-00773-t004:** Live bacteria of two bacteria strains (*Streptococcus sanguinis* and *Lactobacillus salivarius*) and complete oral plaque at 37 °C for 1, 2 and 4 weeks (mean ± standard deviation (SD)).

Strain	Sample	1 Week	2 Weeks	4 Weeks
*S. sanguinis*	Ti	32.0 × 10^4^ ± 2.1 × 10^4^	35.8 × 10^4^ ± 1.9 × 10^4^	42.0 × 10^4^ ± 2.3 × 10^4^
	Ti–N–TSP	1.2 × 10^4^ ± 4 × 10^3^	2.1 × 10^4^ ± 7 × 10^3^	1.9 × 10^4^ ± 3 × 10^3^
*L. salivarius*	Ti	12.5 × 10^4^ ± 1.1 × 10^4^	22.7 × 10^4^ ± 1.9 × 10^4^	32.0 × 10^4^ ± 2.3 × 10^4^
	Ti–N–TSP	8 × 10^3^ ± 1 × 10^3^	2.3 × 10^4^ ± 5 × 10^3^	1.9 × 10^4^ ± 3 × 10^3^
Oral plaque	Ti	31.0 × 10^4^ ± 2.1 × 10^4^	33.6 × 10^4^ ± 1.5 × 10^4^	39.0 × 10^4^ ± 2.0 × 10^4^
	Ti–N–TSP	1.8 × 10^4^ ± 3 × 10^3^	2.9 × 10^4^ ± 6 × 10^3^	2.7 × 10^4^ ± 4 × 10^3^

**Table 5 polymers-12-00773-t005:** Ratio of dead/live bacteria [red cells/(red cells + green cells)] for the different times of incubation at 37 °C (mean ± standard deviation (SD)).

Incubation Time	Sample	*S. sanguinis*	*L. salivarius*	Oral Plaque
1 week	Ti	0.001 ± 0.00	0.003 ± 0.001	0.001 ± 0.00
	Ti–N–TSP	0.17 ± 0.07	0.19 ± 0.08	0.19 ± 0.08
2 weeks	Ti	0.002 ± 0.001	0.04 ± 0.04	0.002 ± 0.002
	Ti–N–TSP	0.16 ± 0.05	0.26 ± 0.08	0.35 ± 0.06
4 weeks	Ti	0.05 ± 0.05	0.03 ± 0.04	0.03 ± 0.03
	Ti–N–TSP	0.39 ± 0.05	0.47 ± 0.20	0.58 ± 0.15

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
