# Peer review of "Antibacterial Properties of Triethoxysilylpropyl Succinic Anhydride Silane (TESPSA) on Titanium Dental Implants"

_polymers, 2020, doi:10.3390/polym12040773_

Round 1

Reviewer 1 Report

In this paper, authors described "Antibacterial properties of triethoxysilylpropyl succinic anhydre silane (TESPSA) onto titanium dental implants". Authors performed physicochemical analysis and carried out several kinds of analysis through in vitro. But there is still an uncompleted issue that in vitro assay. Please see the attached comment. If this article can be published in this journal, authors have to get more specific data after major revision.

1) The schematic illustration of this manuscript (Fig. 1) should be improved. For example, surface treatment steps should represent step by step.

2) Cell viability study should carry out at least 7 days of culture. 1 day is not enough to verify cell cytotoxicity.

3) Biological cell adhesion should show like Fig. 5.

Author Response

Dear Reviewer,

Thanks for taking the time to review our manuscript and suggest to us to improve our work by providing a lot more detail. We have done so, and we are now submitting a manuscript that not only addresses the points the you specifically raised but also many others that we have considered in order to deliver what we think is a much improved version of our work. This version includes a lot more paragraphs in all main sections, restructured subsections, we have improved tables and figures, and a new title to better reflect the contents of our contribution. There are large number of changes and so, we have not specifically highlighted all of them.

We are looking forward to your comments.

Sincerely,

Francisco-Javier Gil Mur

Reviewer 1

Reviewer #1: In this paper, authors described "Antibacterial properties of triethoxysilylpropyl succinic anhydre silane (TESPSA) onto titanium dental implants". Authors performed physicochemical analysis and carried out several kinds of analysis through in vitro. But there is still an uncompleted issue that in vitro assay. Please see the attached comment. If this article can be published in this journal, authors have to get more specific data after major revision.

1) The schematic illustration of this manuscript (Fig. 1) should be improved. For example, surface treatment steps should represent step by step.

According to this suggestion, the surface treatment has been described in more detail in Figure 1, by describing all the treatment steps one by one.

2) Cell viability study should carry out at least 7 days of culture. 1 day is not enough to verify cell cytotoxicity.

In our study, the cell viability was studied according the ISO 10993-5. Briefly, samples were immersed in medium for 72 hours, which allows the release of the toxic chemicals from the surface to the medium. Thus, we believe that 24 hours in contact with cells represent a good time point to study the cell viability.  Cell response with similar coatings has been reported in previous publications both in vitro and in vivo (see Godoy-Gallardo M, Guillem-Marti J, Sevilla P, Manero JM, Gil FJ, Rodriguez D. Mater Sci Eng C Mater Biol Appl 2016;59:524–32. And Godoy-Gallardo M, Manzanares-Céspedes MC, Sevilla P, Nart J, Manzanares N, Manero JM, Gil FJ, Rodriguez D. Mater Sci Eng C 2016;69:538–45).

In addition, in the revised version of the manuscript a new cell adhesion assay has been included (see question 3, reviewer 1). The results exhibits the same trend as obtained in the cell viability experiment (Reviewer 1, question 3). Therefore, we kindly ask the reviewer to accept it as originally written. 

3) Biological cell adhesion should show like Fig. 5

According to the reviewer comment, a new figure (Figure 4b) has been included in the manuscript showing the cell adhesion of HFFs after 24 and 48 hours of incubation. Moreover, detailed information of the experiment has been included in the revised version of the manuscript (see Materials and Methods, section 2.4.2, Results, section 3.2., Discussion and Figure 4b).

Reviewer 2 Report

The article is about the antibacterial properties of TESPSA onto titanium substrate.

  1. I check the paper similarity and it has 46% similarity. It’s not acceptable. (attached file)
  2. Add some quantitative data to the abstract and main mechanism of the antibacterial activity to attract potential readers to read your article and use it.
  3. Select the proper keywords
  4. The introduction is not satisfactory. The authors write two paragraphs with 10 references. The introduction should be rewritten again.
  5. References are out of date. Use the recent articles for the introduction.
  6. In sample preparation, ref 12 is irrelevant.
  7. Mention figure 1 within the text.
  8. Modify figure 1 and draw the step by step of the modification.
  9. The authors use ref 10, 14,15,16,17 to discuss the OWRK equation but these refs do not discuss it.
  10. Reconsider all references. The irrelevant references are so much in the article.
  11. Line 167, it should be fig. 2ai.
  12. Line 168, it should be fig. 2aii.
  13. In cell cytotoxicity, authors just study day 1, It should be measured at day 3 and day 5.
  14. Discussion is too short (it is not a discussion), discuss the result.
  15. Add some quantitative data to the conclusion

Author Response

Dear Reviewer,

Thanks for taking the time to review our manuscript and suggest to us to improve our work by providing a lot more detail. We have done so, and we are now submitting a manuscript that not only addresses the points the you specifically raised but also many others that we have considered in order to deliver what we think is a much improved version of our work. This version includes a lot more paragraphs in all main sections, restructured subsections, we have improved tables and figures, and a new title to better reflect the contents of our contribution. There are large number of changes and so, we have not specifically highlighted all of them.

We are looking forward to your comments.

Sincerely,

Francisco-Javier Gil Mur

Reviewer 2

Reviewer #2: The article is about the antibacterial properties of TESPSA onto titanium substrate.

1) I check the paper similarity and it has 46% similarity. It’s not acceptable. (Attached file)

The reviewer has raised a critical point. To address this important question the manuscript has been revised and modified according the file attached. We apologize for the inconveniences.

2) Add some quantitative data to the abstract and main mechanism of the antibacterial activity to attract potential readers to read your article and use it.

According to the reviewer comment, the abstract has been revised and improved, and quantitative data have been included.

3) Select the proper keywords

New keywords have been introduced in the revised version.

4) The introduction is not satisfactory. The authors write two paragraphs with 10 references. The introduction should be rewritten again. References are out of date. Use the recent articles for the introduction.

The introduction has been rewritten and improved. More recent and relevant articles are cited in the introduction. Thank you for noting that.

5) In sample preparation, ref 12 is irrelevant.  The authors use ref 10, 14,15,16,17 to discuss the OWRK equation but these refs do not discuss it. Reconsider all references. The irrelevant references are so much in the article.

According to the reviewer comment, all references in the manuscript have been checked, deleted or replaced.

6) Mention figure 1 within the text. Modify figure 1 and draw the step by step of the modification.

As commented in previous points (Reviewer 1, question 1) the Figure 1 has been revised and improved and now includes the surface treatment steps.

9) Line 167, it should be fig. 2ai. Line 168, it should be fig. 2aii.

References to the figure have been modified accordingly.

10) In cell cytotoxicity, authors just study day 1, It should be measured at day 3 and day 5.

This question has been commented above (see Reviewer 1, question 2). However, following this comment and the suggestion of Reviewer 1 (see above), complementary studies of cell adhesion have been performed and are now included in the new version (see Materials and Methods, section 2.4.2, Results, section 3.2., Discussion and Figure 4b). We believe the issue of cell viability is better addressed now.

11) Discussion is too short (it is not a discussion), discuss the result.

According to the reviewer’s comment, discussion has been rewritten in order to include a proper results discussion.

12) Add some quantitative data to the conclusion

The reviewer is right, the former conclusion was too general. We have included quantitative data to it in order to improve its clarity.

Round 2

Reviewer 1 Report

This reviewer approves to publish this manuscript as an article.

Reviewer 2 Report

The authors address all comments, and the article is recommended to be published after minor revision about the software error within the text at line 217, 219, 221.